# Sex Hormone-Binding Globulin (SHBG) Reduction: The Alarm Bell for the Risk of Non-Alcoholic Fatty Liver Disease in Adolescents with Polycystic Ovary Syndrome

**DOI:** 10.3390/children9111748

**Published:** 2022-11-15

**Authors:** Flavia Urbano, Mariangela Chiarito, Crescenza Lattanzio, Angela Messa, Marco Ferrante, Mariantonietta Francavilla, Irsida Mehmeti, Giuseppe Lassandro, Paola Giordano, Maria Felicia Faienza

**Affiliations:** 1Pediatric Unit, Giovanni XXIII Pediatric Hospital, 70126 Bari, Italy; 2Unit of Radiology, Giovanni XXIII Pediatric Hospital, 70126 Bari, Italy; 3Faculty of Pharmacy, Catholic University “Our Lady of Good Counsel”, 1000 Tirana, Albania; 4Pediatric Unit, Department of Interdisciplinary Medicine, University of Bari “A. Moro”, 70124 Bari, Italy; 5Pediatric Unit, Department of Precision and Regenerative Medicine and Ionian Area, University of Bari “A. Moro”, 70124 Bari, Italy

**Keywords:** PCOS, NAFLD, adolescents, SHBG, insulin resistance

## Abstract

Polycystic ovary syndrome (PCOS) represents an endocrine condition affecting 5–18% of adolescents, frequently in association with obesity, metabolic alterations, and liver dysfunction. In this study, we aimed to evaluate the prevalence and risk factors for developing non-alcoholic fatty liver disease (NAFLD) in a cohort of PCOS adolescents. Thirty-two girls were assessed for anthropometric and biochemical markers: total cholesterol (TC), high density lipoprotein cholesterol (HDL-C), low density lipoprotein cholesterol (LDL-C), triglycerides (TG), glucose, insulin, alanine aminotransferase (ALT), aspartate aminotransferase (AST) and gamma glutamyl transpeptidase (γGT). In addition, LH, FSH, 17β-Estradiol (E2), prolactin, testosterone (T), free testosterone, delta 4-androstenedione (D4 A), dehydroepiandrosterone sulfate (DHEAS) and sex hormone binding protein (SHBG) were also evaluated. All subjects underwent liver ultrasound to detect NAFLD. Our data demonstrated that PCOS adolescents complicated with NAFLD accounted for 37.5%, and those with obesity and lower SHBG were more predisposed to developing NAFLD. Moreover, SHBG showed a negative correlation with several parameters such as blood pressure, body mass index, waist circumference, insulin, and the homeostatic model assessment of insulin resistance (HOMA-IR). Our results demonstrated that the assessment of SHBG may allow the identification of PCOS adolescents at risk for developing NAFLD and metabolic alterations.

## 1. Introduction

Polycystic ovary syndrome (PCOS) represents an endocrine disorder affecting from 5% to 18% of women of reproductive age [1]. The main characteristics of PCOS are abnormal menstrual cycles (oligomenorrhea or amenorrhea), clinical and/or biochemical hyperandrogenism and the presence or absence of polycystic ovaries [2]. Pelvic ultrasound is not diagnostic of PCOS in adolescents because of the physiological presence of multifollicular ovaries [2]. Currently, the criteria to diagnose PCOS in adolescents are represented by hyperandrogenism (persistent elevation of serum testosterone, moderate–severe hirsutism or acne vulgaris) and oligo-anovulatory cycles (abnormal menstrual pattern and persistent symptoms for 1–2 years) [3,4,5].

PCOS is multifactorial and its etiology is not clearly understood. It is connected with an interplay of genetic and environmental factors. Ovarian hyperandrogenism is the essence of PCOS. It is related to insulin resistance and compensatory hyperinsulinism, which in turn augments the action of luteinizing hormone (LH) on the theca cells, increasing androgen synthesis and reducing hepatic synthesis of sex-hormone-binding globulin (SHBG). Abnormal gonadotropin-releasing hormone (GnRH) regulation, increased LH and decreased follicle-stimulating hormone (FSH) promote the excess of ovarian androgens and ovulatory disfunction. Obesity is involved in the pathophysiology of PCOS by promoting insulin resistance and increasing the hyperandrogenism. Nearly 30–40% of PCOS adolescents are overweight or obese [6]. Although obesity and insulin resistance confer a high risk of further cardiometabolic alterations, they are not considered diagnostic criteria for PCOS [7,8,9,10]. Non-alcoholic fatty liver disease (NAFLD) is a term used to describe a spectrum of hepatic manifestation, ranging from simple hepatic steatosis to non-alcoholic steatohepatitis (NASH), associated with hepatic inflammation and fibrosis. PCOS and NAFLD share many risk factors, especially obesity and insulin resistance. Furthermore, hyperuricemia, which is common in subjects with obesity, metabolic syndrome, and type 2 diabetes, can also predispose to hepatic steatosis, insulin resistance and cardiovascular risk [11,12,13,14]. 

An increased prevalence of NAFLD has been observed in PCOS women with comparable body mass index (BMI) [15,16]. In addition to BMI and the dysregulation of glucose metabolism, the hyperandrogenism is also involved in the development of NAFLD as an independent risk factor. Indeed, PCOS women with androgen excess have higher liver fat accumulation with respect to those with normal androgen levels [15,16,17]. 

Sex hormone binding globulin (SHBG) is a hepatic glycoprotein involved in the transport and regulation of circulating androgen levels. Recently, SHBG has been recognized as a biomarker for several diseases [18]. In particular, low SHBG serum levels have been associated with high levels of markers of inflammation [19], and with the risk of onset and progression of obesity, NAFLD and metabolic syndrome (MetS) [20]. Conversely, the re-establishment of normal SHBG levels has been inversely correlated with a reduction of hepatic fat accumulation [20]. Remarkably, subjects affected with PCOS show high intrahepatic levels and low circulating levels of SHBG [21]. This observation has been confirmed by Simons et al. [22], who reported an inverse association between de novo lipogenesis and SHBG, which may account for the decreased SHBG levels observed in the obese subjects. Reducing hepatic SHBG production may represent a crucial step in the pathogenesis of PCOS, so this hepatokine could be considered an early biomarker of PCOS [23]. In addition, low SHBG levels have been linked with increased insulin resistance, although the exact mechanism connecting insulin resistance and SHBG suppression remains unclear [24]. Notably, in subjects affected with NAFLD, insulin resistance has an important role in hepatic de novo lipogenesis [24]. Therefore, the close interplay between SHBG levels and risk factors involved in the pathogenesis of NAFLD, obesity and MetS suggests SHBG as a potential therapeutic target in the management of these diseases [25]. 

Recently, a correlation between SHBG and NAFLD has been observed in adult women with PCOS [26]. There are no data about the risk of developing NAFLD in adolescents affected with PCOS, or about the role of SHBG.

The aim of this study was to investigate the prevalence of NAFLD in a cohort of obese and non-obese adolescents with PCOS and to evaluate the role of hyperandrogenism as an independent risk factor in the development of NAFLD.

## 2. Materials and Methods

### 2.1. Study Population

Thirty-two adolescents aged 11.64 to 19.83 years (median age: 15.86 years) diagnosed with PCOS who were referred to the Endocrinology Unit of the Pediatric Clinic of University of Bari (Italy) were enrolled for this study. Inclusion criteria were: Caucasian race, age ranging from 10 to 20 years and pubertal development according to Tanner stage IV or V. Exclusion criteria were: chronic liver diseases, chronic systemic diseases and medications (including contraceptives). Informed consent was obtained from all subjects or their parents. The study was conducted in accordance with the Declaration of Helsinki, and the protocol was approved by the Ethics Committee of Policlinico of Bari (Project identification code 7051, approval date: 10 November 2021).

### 2.2. Diagnosis of PCOS

The diagnosis of PCOS was made based on the presence of an irregular menstrual cycle (oligomenorrhea or amenorrhea) and clinical and/or biochemical hyperandrogenism. Oligomenorrhea was defined differently according to the post-menarche time: up to one year post-menarche, cycles > 60 days; from one to three years after menarche, cycles < 21 or >45 days; more than three years after menarche, cycles > 38 days or less than nine cycles per year [5]. Amenorrhea was defined as the total absence of menses for more than 90 days in previously menstruating girls. Clinical hyperandrogenism was defined by the presence of acne and/or hirsutism assessed by Ferriman and Gallwey score. Biochemical hyperandrogenism was identified by serum elevation of free testosterone. Furthermore, possible differential diagnoses, such as congenital adrenal hyperplasia, including non-classic adrenal hyperplasia and Cushing’s syndrome, were excluded for all participants.

### 2.3. Clinical Data Collection

A detailed anamnesis and a careful physical examination were performed. Body weight (W) and height (H) were measured using standard techniques. BMI was calculated using the formula W/H2 (kg/m^2^). Auxological data were expressed in standard deviation scores (SDSs) for age and gender according to the Italian growth standards [27]. Waist circumference (WC) and the ratio of WC to height (H) (WC/H) were also calculated. Systolic (SBP) and diastolic (DBP) blood pressure were also measured.

### 2.4. Laboratory Data Collection

Total cholesterol (TC), high density lipoprotein cholesterol (HDL-C), low density lipoprotein cholesterol (LDL-C), triglycerides (TG), glucose, insulin, alanine aminotransferase (ALT), aspartate aminotransferase (AST) and gamma glutamyl transpeptidase (γGT) were assessed after an overnight fast. Insulin resistance was evaluated by the homeostatic model assessment of insulin resistance (HOMA-IR), calculated by the formula fasting glucose (mg/dL) × insulin (µU/mL)/405 and the glucose (mg/dL) to insulin (µU/mL) ratio. Adolescents with a family history of diabetes underwent an oral glucose tolerance test (OGTT). Moreover, LH, FSH, 17β-Estradiol (E2), prolactin, testosterone (T), free testosterone, delta 4-androstenedione, dehydroepiandrosterone sulfate (DHEAS) and SHBG were measured.

### 2.5. Imaging Studies Data Collections

Non-alcoholic fatty liver disease (NAFLD) was assessed by liver ultrasound, and performed by an experienced pediatric radiologist, using a convex probe with a frequency between 2.5 and 4 MHz (ultrasound machine Logiq E9, General Electric HealthCare, Chicago, Illinois, USA). The radiologist was unaware of the clinical course and laboratory details of the patients. All measurements were based on sagittal, transverse and oblique scans in which both liver parenchyma and right kidney were simultaneously imaged during maximal inspiration. Non-steatosic hepatic parenchyma displays an echotexture like that of renal parenchyma, but when infiltrated with fat it appears ‘brighter’ [28]. This hepatorenal contrast can be used to detect hepatosteatosis. The presence of steatosis was classified as: grade 1—mild (liver echogenicity slightly increased compared to that of the kidney, with normal visualization of diaphragm and intrahepatic vessel borders), grade 2—moderate (more pronounced difference between liver and kidney, and slightly impaired visualization of intrahepatic vessels and diaphragm), grade 3—severe (hepatic echogenicity significantly increased compared to the kidney, with poor visualization of intrahepatic vessel borders, diaphragm and posterior segments of right hepatic lobe) [29].

### 2.6. Statistical Analysis 

Data were expressed as mean ± SD when normally distributed and as median (quartile) for parameters with non-normal distribution, unless otherwise specified; categorical variables were reported as number and percentage. Subjects were subdivided according to presence/absence of steatosis, and comparisons were performed between groups using Student’s test for normally distributed data and the Mann–Whitney U test for not normally distributed data. Correlations were assessed using the Pearson or the Spearman method. Logistic regression was used, to investigate risk factors for NAFLD. Results are presented in β regression coefficients and 95% confidence intervals. Statistical analysis was performed with the statistical package SPSS v23 for Windows (SPSS Inc., Chicago, IL, USA) and a probability value of *p* < 0.05 was considered statistically significant. Sample size was calculated according to the following formula: 1.962 × Patt × (1 − Patt)/D2 (Patt: expected prevalence, D: power of the study).

## 3. Results

### 3.1. Prevalence of NAFLD 

A total of 12 out of 32 (37.5%) PCOS adolescents were diagnosed with NAFLD. In detail, five (15.6%) were classified as having grade 1 (mild steatosis), four (9.4%) as having grade 2 (moderate steatosis) and three (12.5%) as having grade 3 (severe steatosis). Among the three patients with severe steatosis, one patient also had an increase in transaminase, configuring the diagnosis of steatohepatitis. Among the 12 PCOS adolescents with steatosis, one was overweight (grade 1) while the others were obese.

The response rate obtained was 100%.

### 3.2. Differences between NAFLD and Non-NAFLD PCOS Adolescents

The main characteristics of the 32 PCOS adolescents enrolled for this study are shown in Table 1. PCOS adolescents affected with NAFLD had significantly higher W, BMI, WC and WC/H, as well as SBP and DBP (*p* < 0.05 for all variables), than the non-NAFLD PCOS adolescents. Moreover, NAFLD PCOS adolescents also had significantly higher serum levels of alanine aminotransferase (ALT), insulin and HOMA-IR than the non-NAFLD adolescents (*p* < 0.05). In addition, the former had significantly lower SHBG serum levels than the latter (*p* < 0.01). No significant differences were observed in other parameters of lipid profile (TC, HDL-C, LDL-C, TG) and hormonal profile (FSH, LH, E, DHEAS, delta 4-androstenedione, T, free T).

### 3.3. Correlations of SHBG with Clinical and Biochemical Parameters 

Given the significant difference in SHBG serum levels between NAFLD and non-NAFLD PCOS adolescents, we performed univariate correlations between SHBG and clinical and biochemical parameters, as shown in Table 2. SHBG was negatively correlated with W, BMI, SBP, DBP, insulin, and HOMA-IR. The same variables were significantly higher in the NAFLD PCOS adolescents, suggesting a possible role of SHBG as a predictor of NAFLD.

### 3.4. Logistic Regression Analysis of Influencing Factors of PCOS Complicated with NAFLD

To define if SHBG can be predictive of NAFLD in PCOS adolescents, a binomial logistic regression was performed, considering NAFLD as the independent variable and SHBG as the covariate. The analysis shows that BMI is a risk factor for NAFLD in girls with PCOS (OR: 1.2, IC: 1.192–1.291), and also that a reduction in SHBG serum levels is associated with higher risk of NAFLD in PCOS adolescents (OR: 0.953, IC: 0.936–0.971, *p* < 0.05).

## 4. Discussion

NAFLD represents a spectrum of progressive liver damage that progressively evolves from mild fatty liver to severe fibrosis. In the initial stages, NAFLD can be reversible, so accurate screening of subjects with risk factors is mandatory to avoid irreversible damages [30]. NAFLD is an important health issue in PCOS subjects, but to date is still little studied, especially in adolescents. Our results showed that the rate of PCOS adolescents complicated with NAFLD was 37.5%, which is in accordance with a previous study in adult PCOS women, reporting a prevalence ranging from 24 to 69% [14]. It has been shown that obesity is one of the most important risk factors for PCOS and more than around 30–40% of PCOS women are obese. We found that NAFLD PCOS adolescents had higher BMI, in line with previous studies [31,32], and higher WC and WC/H than the non-NAFLD PCOS adolescents. In accordance with the severity of steatosis, these variables were higher in the most severe phenotypes, suggesting that obesity and visceral adiposity are closely related to PCOS complicated with NAFLD.

Previous studies have shown that high androgen levels may play a role in the increased rate of NAFLD in women with PCOS [15,17,33,34]. Data in vitro showed that androgens can affect hepatic liver metabolism in a dimorphic way according to sex; indeed, androgens can potentiate lipogenic gene expression and de novo lipogenesis in human hepatocytes in females, but not in males [35]. The different effects that androgens exert depending on sex is an emerging and interesting topic, female androgen excess and male hypogonadism sharing an overlapping metabolic phenotype characterized by abdominal obesity, dyslipidemia, insulin resistance, and NAFLD [36].

However, there is a lack of evidence for the association between hyperandrogenism and NAFLD in adolescents. In our study, we did not find any statistically significant difference either in total or free testosterone between NAFLD and non-NAFLD PCOS adolescents. This finding is in accordance with the study of Cree-Green’s that reported a prevalence of 49% of NAFLD in PCOS compared to 14% in a healthy control group, without any correlation with serum androgen levels [37]. Interestingly, we found reduced serum levels of SHBG in our NAFLD PCOS adolescents. This reduction in SHBG levels could increase the risk of NAFLD in PCOS adolescents. Moreover, in our cohort, obese adolescents had lower SHBG serum levels with respect to both those overweight and normal weight. These data are in accordance with the study of Di Stasi et al. who found SHBG as a biomarker of NAFLD in a cohort of 66 PCOS women [26]. Moreover, Mueller et al., in a cohort of 573 children with liver biopsy, found that SHBG was inversely associated with the histologic severity of steatosis [38].

By analyzing the correlations between SHBG and several parameters, negative correlations were found with W, BMI, blood pressure, insulin and HOMA-IR. These results confirmed previous data reporting SHBG as an indicator of liver and metabolic impairment [35]. The role of SHBG as a marker of metabolic alterations is not fully understood; indeed, it can be a cause or a consequence of liver dysfunction. In vitro studies showed that SHBG reduces inflammation and storage of lipid in macrophages and adipocytes, which can explain the protective effect of SHBG versus metabolic syndrome and its complications [19]. These protective effects could be mediated through PPARγ modulation [39]. However, SHBG production is downregulated by an increased hepatic lipogenesis and pro-inflammatory cytokines secretion [40]. As, in our study, we found that the NAFLD PCOS adolescents had a higher BMI than non-NAFLD PCOS adolescents, we can hypothesize that the development of obesity in girls with PCOS further reduced SHBG levels, which in turn have triggered the development of NAFLD.

To our knowledge, this is the first study which investigated NAFLD in a cohort of adolescents with PCOS, and this represents the strength of this work. The relevance of our results relies on the recognition of two risk factors such as high BMI and low SHBG for NAFLD development in PCOS adolescents. From this perspective, weight loss and a healthier lifestyle must be always encouraged. An ultrasound examination of the liver should be performed regularly in PCOS girls, for early detection of liver steatosis. SHBG could represent the first “alarm bell” to deepen liver conditions in PCOS patients.

Further studies will be needed to evaluate the role of other factors, such as pro-inflammatory (e.g., TNFα, IL6) and anti-inflammatory (e.g., adiponectin) cytokines, in linking the PCOS with the risk of developing NAFLD.

## Figures and Tables

**Table 1 children-09-01748-t001:** General characteristics of NAFLD and non-NAFLD adolescents.

	NAFLD Adolescents	Non-NAFLD Adolescents	*p* Value
Age	15.19 ± 2.42	16.2 ± 2	n.s
Height (SDS)	0.22 (−0.14–0.68)	−0.65 (−0.95–0.18)	n.s
Weight (Kg)	94.83 ± 21.85	63.93 ± 14.78	<0.05
BMI (kg/m^2^)	35.31 ± 7.9	25.76 ± 6.28	<0.05
WC (cm)	107.75 (95–112)	76.5 (67–91)	<0.05
WC/H	0.63 ± 0.10	0.51 ± 0.09	<0.05
SBP (mmHg)	130 (115–137.5)	120 (105–125)	<0.05
DBP (mmHg)	81.25 ± 11.1	72.37 ± 9.77	<0.05
Total cholesterol (mg/dL)	160.5 (147–179)	138 (123–173.5)	n.s
LDL-cholesterol (mg/dL)	92.27 ± 25.37	80 ± 23.75	n.s
HDL-cholesterol (mg/dL)	53 ± 11.09	55.33 ± 12.78	n.s
Triglycerides (mg/dL)	89.81 ± 63.56	67.14 ± 42.96	n.s
AST (U/L)	23.10 ± 11.78	18.57 ± 4.40	n.s
ALT (U/L)	27 (21–37.5)	20.5 (18–25.5)	n.s
FSH (mUI/mL)	6.84 (4.49–8.2)	3.82 (3.24–6.96)	n.s
LH (mUI/mL)	14.4 (6.92–16.9)	6.84 (4.44–15)	n.s
Total Testosterone (ng/mL)	0.45 ± 0.19	0.43 ± 0.24	n.s
Free Testosterone (pg/mL)	2.4 ± 0.78	2.08 ± 1.02	n.s
DHEAS (ug/dL)	308 ± 126.63	305.45 ± 150.38	n.s
Δ4-androstenedione (ng/dL)	1.75 ± 0.89	1.62 ± 0.73	n.s
17β-Estradiol	37.25 (26.45–43.6)	60.05 (37.75–90.85)	n.s
Insulin (microUI/mL)	27 (12.6–35.2)	10.5 (8.45–21.4)	<0.05
HOMA-IR	6.07 (2.61–7.3)	2.12 (1.72–4.51)	<0.05
SHBG (nmol/L)	16.65 (13.25–20.65)	37.95 (22.8–46.1)	<0.01

Data are expressed as mean ± SD when normally distributed, median (quartile) when not normally distributed. Differences in not normally distributed continuous variables were assessed by Mann–Whitney U test for comparison between the two groups. Differences in normally distributed continuous variables were assessed by unpaired *t* test. n.s: not statistically significant. BMI: body mass index; WC: waist circumference; WC/H: waist circumference/height; SBP: systolic blood pressure; DBP: diastolic blood pressure; ALT: alanine aminotransferase, AST: aspartate aminotransferase; FSH: follicle stimulating hormone; LH: luteinizing hormone; DHEAS: dehydroepiandrosterone sulfate; HOMA-IR: homeostatic model assessment of insulin resistance; SHBG: sex hormone binding globulin.

**Table 2 children-09-01748-t002:** Univariate correlations between SHBG and clinical and biochemical parameters.

	SHBG	
	R	*p* Value
Weight (kg)	−0.612	<0.05
BMI (kg/m^2^)	−0.677	<0.05
SBD (mmHg)	−0.416	<0.05
DBP (mmHg)	−0.399	<0.05
Insulin (microUI/mL)	−0.595	<0.05
HOMA-IR	−0.607	<0.05

BMI: body mass index; SBP: systolic blood pressure; DBP: diastolic blood pressure; HOMA-IR: homeostatic model assessment of insulin resistance.

## Data Availability

The data presented in this study are available on request from the corresponding author.

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
