# Peer review of "Sex Hormone-Binding Globulin (SHBG) Reduction: The Alarm Bell for the Risk of Non-Alcoholic Fatty Liver Disease in Adolescents with Polycystic Ovary Syndrome"

_children, 2022, doi:10.3390/children9111748_

Round 1

Reviewer 1 Report

Dear Editor,

Thanks for giving me this opportunity to review the manuscript children-1968976. In this paper, the authors investigated the “Sex Hormone-Binding Globulin (Shbg) Reduction: The Alarm Bell for the Risk of Non-Alcoholic Fatty Liver Disease in Adolescents with Pcos”.

This study aimed to determine the association between SHBG with NFLD.

Recommendation: In my opinion the above mentioned article have sufficient novelty, however need minor revision which help to enhance the quality of paper.

My specific comments are as follows:

Please edit the word “Pcos” in the title, don’t write the abbreviation in the title

How sample size was calculated in this study? Please add the formula used for calculating it.

It would be better to have a healthy control group.

Please add the response rate in the result section.

The manuscript must be edited by a native speaker.

What is the clinical implication of this study?

Please add the limitation and strengths of the study with more details. 

Author Response

1.Please edit the word “PCOS”, dont’write the abbreviation in the title

Answer: we edited in the title the term PCOS.

  1. How the sample size was calculated in this study? Please add the formula used for calculating it

Answer: we added the formula used for calculating sample size.

  1. It would be better to have a healthy control group

Answer: we thank the reviewer for his/her suggestion. However, this study is a cohort study in which we aimed to correlate the hyperandrogenism that characterizes the PCOS adolescents with the risk for developing NAFLD. We choosed don’t include a control group because the hyperandrogenism is not present in healthy adolescents.

  1. Please add the response rate in the result section.

Answer: we added this information in the result section.

  1. The manuscript must be edited by a native speaker

Answer: the manuscript has been edited by native english speaker.

  1. Please add the limitation and the strenght of the study with more detail

Answer: we added limitation and strenght.

Reviewer 2 Report

The manuscript by Urbano et al. titled “SEX HORMONE-BINDING GLOBULIN (SHBG) REDUCTION: THE ALARM BELL FOR THE RISK OF NON-ALCOHOLIC FATTY LIVER DISEASE IN ADOLESCENTS WITH PCOS” studied the prevalence and the risk factors for non-alcoholic fatty liver disease (NAFLD) in a cohort of PCOS adolescents. The manuscript is interesting and technically sound; however, this reviewer has some concerns and suggestions that need to be answered that preclude its further consideration for publication.

Major Concerns:

1   1.  Introduction is correct in terms of defining PCOS and NAFLD. However, the reader will benefit in this section the inclusion of information regarding SHBG, its levels and regulation in obesity, PCOS and NAFLD in adults and children.

2   2.  Pro-inflammatory (e.g. TNFα, IL6) and anti-inflammatory (e.g. adiponectin) factors should be measured. The results of these measurements could give important information of why PCOS adolescents have NAFLD. This will increase the importance and relevance of this study.

Minor Concerns:

1   Introduction section: it has been demonstrated that insulin does not regulate hepatic SHBG production although there is a clear correlation between insulin and SHBG for instance in obesity. There is an important amount of literature describing it in human studies, animal models and cells.

Author Response

  1. Introduction is correct in terms of defining PCOS and NAFLD. However, the reader will benefit in this section the inclusion of information regarding SHBG, its levels and regulation in obesity, PCOS and NAFLD in adults and children.

Answer: we added in the introduction the informations about SHBG and its link with obesity, NAFLD and PCOS.

2   Pro-inflammatory (e.g. TNFα, IL6) and anti-inflammatory (e.g. adiponectin) factors should be measured. The results of these measurements could give important information of why PCOS adolescents have NAFLD. This will increase the importance and relevance of this study.

Answer: we really thank to the referee for his/her suggestion. We did not perform the assesment of these cytokines in this study but it will be a good suggestion for a future study. We reported this as limitation of our study.

Minor Concerns:

  • Introduction section: it has been demonstrated that insulin does not regulate hepatic SHBG production although there is a clear correlation between insulin and SHBG for instance in obesity. There is an important amount of literature describing it in human studies, animal models and cells.

Answer: we added this issue in the introduction section (ref.19)

Round 2

Reviewer 2 Report

The manuscript by Urbano et al. titled “SEX HORMONE-BINDING GLOBULIN (SHBG) REDUCTION: THE ALARM BELL FOR THE RISK OF NON-ALCOHOLIC FATTY LIVER DISEASE IN ADOLESCENTS WITH PCOS” studied the prevalence and the risk factors for non-alcoholic fatty liver disease (NAFLD) in a cohort of PCOS adolescents.

The authors answered some of the concerns raised by this reviewer but the manuscript can be improved before considering its publication. The Introduction and Discussion sections of this manuscript are rather short and both sections can still be improved considerably:

Authors need to understand that their results are relevant and it is very important to highlight in the Introduction section the relationship between SHBG and Obesity and NAFLD, including not only the molecular pathways regulating SHBG in Obesity and NAFLD but also the role of SHBG in the progression of these diseases. This information is relevant since only the obese/overweight girls with PCOS develop NAFLD. Discussion section should consider this information to debate whether or not the development of Obesity in PCOS girls is further reducing SHBG levels which in turn triggers the development of NAFLD.

Author Response

The authors answered some of the concerns raised by this reviewer but the manuscript can be improved before considering its publication. The Introduction and Discussion sections of this manuscript are rather short and both sections can still be improved considerably:

Authors need to understand that their results are relevant and it is very important to highlight in the Introduction section the relationship between SHBG and Obesity and NAFLD, including not only the molecular pathways regulating SHBG in Obesity and NAFLD but also the role of SHBG in the progression of these diseases. This information is relevant since only the obese/overweight girls with PCOS develop NAFLD. Discussion section should consider this information to debate whether or not the development of Obesity in PCOS girls is further reducing SHBG levels which in turn triggers the development of NAFLD.

Answer: we really thank to the reviewer for his/her suggestions that allow us to improve our manuscript.
We added in the introduction the information about the link between obesity, NAFLD and SHBG and we reported new references (from 18 to 25) and we discussed this in relation of our results.
